# Systemic and Peripheral Mechanisms of Cortical Stimulation-Induced Analgesia and Refractoriness in a Rat Model of Neuropathic Pain

**DOI:** 10.3390/ijms24097796

**Published:** 2023-04-25

**Authors:** Danielle V. Assis, Ana Carolina P. Campos, Amanda F. N. Paschoa, Talita F. Santos, Erich T. Fonoff, Rosana L. Pagano

**Affiliations:** 1Laboratory of Neuroscience, Hospital Sírio-Libanês, São Paulo 01308-060, SP, Brazil; 2Division of Functional Neurosurgery, Department of Neurology, University of Sao Paulo Medical School, São Paulo 05402-000, SP, Brazil

**Keywords:** motor cortex stimulation, neuropathic pain, sciatic nerve, inflammation, neurotrophins, substance P, β-endorphin

## Abstract

Epidural motor cortex stimulation (MCS) is an effective treatment for refractory neuropathic pain; however, some individuals are unresponsive. In this study, we correlated the effectiveness of MCS and refractoriness with the expression of cytokines, neurotrophins, and nociceptive mediators in the dorsal root ganglion (DRG), sciatic nerve, and plasma of rats with sciatic neuropathy. MCS inhibited hyperalgesia and allodynia in two-thirds of the animals (responsive group), and one-third did not respond (refractory group). Chronic constriction injury (CCI) increased IL-1β in the nerve and DRG, inhibited IL-4, IL-10, and IL-17A in the nerve, decreased β-endorphin, and enhanced substance P in the plasma, compared to the control. Responsive animals showed decreased NGF and increased IL-6 in the nerve, accompanied by restoration of local IL-10 and IL-17A and systemic β-endorphin. Refractory animals showed increased TNF-α and decreased IFNγ in the nerve, along with decreased TNF-α and IL-17A in the DRG, maintaining low levels of systemic β-endorphin. Our findings suggest that the effectiveness of MCS depends on local control of inflammatory and neurotrophic changes, accompanied by recovery of the opioidergic system observed in neuropathic conditions. So, understanding the refractoriness to MCS may guide an improvement in the efficacy of the technique, thus benefiting patients with persistent neuropathic pain.

## 1. Introduction

Neuropathic pain (NP) is a painful condition resulting from an injury or disease that affects the somatosensory nervous system [1,2]. It is maladaptive and loses its protective function, becoming a self-perpetuating pathological symptom [3]. Approximately 7–8% of adults in the general population suffer from chronic NP [4], and due to its multifactorial and complex pathophysiology, only 30 to 40% of patients benefit from the use of conventional pharmacotherapy [5,6,7], making the NP a challenging condition to understand and treat. Consequently, this disease is commonly related to poor quality of life [8]; therefore, the comprehension of its treatment is a form of public health measure and needs careful attention.

Peripheral and central sensitization phenomena occur in peripheral NP after chronic nerve injury, characterized by synaptic plasticity, glial activation, and neuronal hyperactivity events, leading to the development and maintenance of NP [9,10]. Peripheral sensitization is, at least partially, induced by the activation of Schwann cells and immune cells, which release algogenic mediators such as nerve growth factor (NGF) and proinflammatory cytokines, together with substance P (SP) released by primary afferent neurons [9,11,12,13]. Peripheral sensitization also involves the activation of satellite glial cells (SGCs) present in the dorsal root ganglion (DRG), which also releases proinflammatory cytokines that contribute to NP [14,15,16]. In the dorsal horn of the spinal cord, primary afferent neurons and other local neurons and glial cells release SP, glutamate, cytokines, and neurotrophic factors, which contribute to central sensitization [17,18], and thus, with the intensity of nociceptive information delivered to the sensory and limbic cortices [19,20].

Considering the complexity of persistent NP, one-third of patients are refractory to conventional treatment, making the search for new therapeutic approaches undoubtedly of great value. In this context, epidural electrical stimulation of the primary motor cortex (MCS), a non-destructive, adjustable, and reversible therapeutic technique [21], has been used in refractory patients and has shown satisfactory outcomes in attenuating NP [22]. MCS can reduce pain in approximately 50% of patients [23,24], reporting an efficacy greater than 50% in pain relief, making it a promising therapy that can improve the functional capacity and quality of life of these patients [25,26,27]. The analgesic efficacy of MCS has been attributed to the modulation of supraspinal areas involved in the perception and/or emotional appraisal of pain, including the anterior cingulate, insular and prefrontal cortices, basal ganglia, and periaqueductal gray matter in individuals [28,29,30,31] and NP preclinical models [32,33,34,35,36]. The underlying mechanisms of MCS-induced analgesia involve the activation of opioid, cannabinoid, dopaminergic, and serotonergic systems, through descending analgesic pathways [33,34,37,38], leading to a decrease in neuronal hyperactivation in the spinal cord [32]. We previously demonstrated a direct correlation between the effectiveness of MCS and the inhibition of tumor necrosis factor (TNF)-α and interleukin (IL)-1β levels in spinal astrocytes and microglia in NP rats [38]. These results shed light on the spinal anti-inflammatory mechanisms involved in cortical stimulation-induced analgesia.

Nevertheless, most studies have focused on understanding MCS-induced analgesia in relation to central mechanisms of painful modulation; however, there has been limited discussion its influence on peripheral modulation. Considering the crucial role of peripheral neuroimmune activation in NP pathophysiology, we hypothesize that MCS analgesic success is directly related to the attenuation of classic algogenic mediators. In an attempt to comprehend the main molecular signature differences between MCS-induced analgesia and MCS refractoriness, regarding the peripheral mechanisms of NP, we aimed to evaluate the main peripheral discrepancies between effective and refractory MCS to evaluate peripheral mediators involved in the control of NP.

## 2. Results

### 2.1. Experimental Design

The animals were evaluated in nociceptive tests (basal measurement), followed by chronic constriction injury (CCI) or false operation (FOP) procedures. After seven days, the animals received transdural electrodes in the cortex motor implants. On the 14th day after CCI, the animals were reevaluated before (intermediate measurement) and after (final measurement) MCS. FOP and CCI animals were also evaluated but not stimulated. After the last nociceptive test, the animals were euthanized, and the tissues (sciatic nerve and DRG) and peripheral blood (plasma) were collected for further analysis of the expression of cytokines, neurotrophic factors (NGF and brain derived neurotrophic factor—BDNF), SP, and β-endorphin. Based on the assessment of nociceptive behavior, the animals were divided into four experimental groups: FOP rats (painless control; *n* = 10); rats with CCI (with NP; *n* = 10); rats with CCI responsive to MCS (with NP reversal; *n* = 16) and rats with CCI refractory to MCS (without NP reversal; *n* = 10) (Figure 1).

### 2.2. Characterization of Effectiveness and Refractoriness to MCS

NP was characterized 14 days after CCI and, as expected, induced mechanical hyperalgesia (F_(2.10)_ = 50.24, *p* < 0.0001; Figure 2A) and mechanical allodynia (F_(2.10)_ = 15.44, *p* < 0.0001; Figure 2B) when compared to FOP animals (control group). The animals were classified into treatment-responsive and treatment-refractory groups based on the MCS response. Considering the response to hyperalgesia, 62% of the stimulated animals showed a reversal of this phenomenon (Figure 2A), characterizing the responsive group to MCS, versus 38% of the animals that did not present this reversion, characterizing the refractory group to MCS (Figure 2). MCS could partially reverse allodynia when compared to the control group (Figure 2B). No changes were observed in the nociceptive threshold of the contralateral paw after CCI in the right posterior paw, or after cortical stimulation.

### 2.3. Expression Pattern of Cytokines and NGF in the Sciatic Nerve

Considering that NP is closely related to peripheral neuroinflammation [11,13,39], and that the inhibition of proinflammatory cytokines and nociceptive factors has been shown to dampen pain maintenance [40], we evaluated classic pro- and anti-inflammatory cytokines and NGF in the sciatic nerve, to better comprehend the effect of MCS-induced analgesia (responsive) or failure (refractory). Fourteen days after neuropathy, as expected, the CCI group showed an increased expression of IL-1β (F_(3.20)_ = 8.860, *p* = 0.0004; Figure 3B) and inhibited expression of IL-4 (F_(3.20)_ = 117.0, *p* < 0.0001; Figure 3C), IL-10 (F_(3.20)_ = 7.666, *p* = 0.0008; Figure 3E), and IL-17A (F_(3.20)_ = 5.091, *p* = 0.0072; Figure 3F) in the sciatic nerve, when compared to the FOP control group. Among the responsive animals, MCS increased the nerve expression of IL-6 (F_(3.20)_ = 8.409, *p* = 0.0006; Figure 3D), restored the IL-10 (Figure 3E) and IL-17A (Figure 3F) levels, and decreased the NGF expression (F_(3,20)_ = 3.666, *p* = 0.0297; Figure 3I) compared to FOP animals. Rats refractory to cortical stimulation showed increased expression of TNF-α (F_(3.20)_ = 2.996, *p* = 0.0482; Figure 3A) and decreased expression of interferon (IFN) γ (F_(3.20)_ = 14.94, *p* = 0.0001; Figure 3G) in the sciatic nerve compared to the other groups. Regarding fractalkine/CX3C chemokine ligand 1 (CX3CL1), no changes in the expression of this chemokine were observed among the different experimental groups evaluated (F_(3.20)_ = 2.872, *p* = 0.0564; Figure 3H).

### 2.4. Expression Pattern of Cytokines in the DRG

To continue the investigation of peripheral neuroinflammation, we evaluated cytokines, classically involved in persistent pain [16,41], in the neuronal bodies cluster found in the DRG. Two weeks after the nerve lesion, the CCI group showed increased expression of IL-1β in the DRG compared to that in the control group (F_(3.20)_ = 7.857, *p* = 0.0036; Figure 4B). MCS per se, regardless of its therapeutic effect, reversed this increase (Figure 4B). For the refractory group, we observed a decreased expression of TNF-α compared to the CCI group (F_(3.20)_ = 4.431, *p* = 0.0218; Figure 4A), and IL-17 compared to the FOP group (F_(3.20)_ = 4.291, *p* = 0.0310; Figure 4F), in the DRG of neuropathic rats. There was no change in the expression of IL-4 (F_(3.20)_ = 0.6559, *p* = 0.5925; Figure 4C), IL-6 (F_(3.20)_ = 0.4308, *p* = 0.7355; Figure 4D), IL-10 (F_(3.20)_ = 0.392, *p* = 0.7607; Figure 4E), IFNγ (F_(3.20)_ = 1.089, *p* = 0.3862; Figure 4G), and CX3CL1 (F_(3.20)_ = 0.8113, *p* = 0.5118; Figure 4H) among the different groups evaluated.

### 2.5. Expression Pattern of SP, β-Endorphin, BDNF, and NGF in the Plasma

For a deeper investigation, we analyzed the circulant levels of factors involved in the NP control [42,43,44] and interestingly, peripheral neuropathy increased circulating SP levels (F_(3.20)_ = 24.63, *p* = 0.0001; Figure 5A) and decreased β-endorphin levels (F_(3.20)_ = 6.762, *p* = 0.0021; Figure 5B) when compared to FOP animals. Regardless of its therapeutic effect, MCS decreased SP (Figure 5A) compared to CCI animals. MCS restored systemic β-endorphin expression only in the responsive group compared to the FOP and CCI animals (Figure 5B). Despite a 69% decrease in circulating BDNF levels after CCI, no statistically significant difference was observed among the different groups evaluated (F_(3.20)_ = 1.47, *p* = 0.2628; Figure 5C). NGF was not detected in plasma samples from all investigated groups. This technical issue may be due to the sensitivity spectrum of the assay, considering the low amount of circulating NGF at the time of collection, possibly as a result of the natural biodegradation process and/or its binding property to different plasma proteins.

### 2.6. Correlations between Nociceptive Threshold, Expression of Cytokines, Neurotrophins, β-Endorphin, and SP with MCS

In order to verify if there is a correlation between the nociceptive threshold (observed with the paw pressure test) and the expression of neurotransmitters in the nerve, DRG, and plasma after cortical stimulation, we correlated the release of cytokines, neurotrophins, β-endorphin, and SP with responsive and refractory rats to MCS. We observed that there is a high negative correlation between the nociceptive threshold and NGF expression in the sciatic nerve in the refractory group, when compared with the responsive group (r^2^ = 0.89, *p* = 0.004, Table 1). We also observed that pain intensity has a high negative correlation with IL-4 expression in the DRG in the refractory group (r^2^ = 0.77, *p* = 0.02, Table 1). For the other analytes, no correlation was observed in the analyzed structures (Table 1).

Correlation analysis between the nociceptive threshold and: (1) expression of TNF-α, IL-1β, IL-4, IL-6, IL-10, IL-17, IFNγ, CX3CL1, and NGF in the sciatic nerve; (2) expression of TNF-α, IL-1β, IL-4, IL-6, IL-10, IL-17, IFNγ, and CX3CL1 in the dorsal root ganglion (DRG); and (3) expression of β-endorphin and SP in plasma, from responsive versus refractory animals to cortical stimulation. The Pearson correlation (r^2^) and significance index (*p*) are calculated. The analytes that showed a mean negative correlation (r^2^ = 0.5 to 0.75) and a high correlation (r^2^ > 0.75) were NGF in the sciatic nerve and IL-4 in the DRG.

## 3. Discussion

In the present study, as expected, neuropathic pain, confirmed by the presence of hyperalgesia and allodynia phenomena, induced peripheral inflammation and systemic sensitization as previously demonstrated in the literature [45,46]. We found that MCS attenuated the nociceptive behavior induced by peripheral neuropathy in 62% of the animals (the responsive group) and failed to induce analgesia in 38% (the refractory group). The analgesic effect of MCS was accompanied by an increase in the pleiotropic cytokine IL-6, restoration of IL-10 and IL-17, and a decrease in NGF in the nerves of neuropathic animals. Furthermore, responsive animals showed restoration of plasma β-endorphin, thus attenuating CCI-induced NP. In contrast, refractory animals showed increased TNF-α expression in the sciatic nerve and failed to modulate inflammation in the DRG, as well as the systemic opioidergic deficit induced by CCI.

MCS is an adjustable and reversible therapeutic technique for treating patients with central or peripheral NP syndromes that are refractory to other types of treatment, with good analgesic response [21]. The accuracy of electrode implantation over the motor cortex somatotopy corresponding to the painful area is essential for cortical stimulation-induced analgesia [47,48]. Hamani et al., in a randomized, double-blind, sham-controlled, single-center trial, compared long-term MCS responsiveness to specific pain etiologies, as a response to the insertion effect during electrode implantation [49]. However, some patients are refractory to this therapeutic option, and there is a need to understand the functional and molecular differences between responsive and refractory analgesia of this neurofunctional intervention.

Consistent with the literature, our results showed that CCI-induced hyperalgesia and allodynia-like behaviors on the ipsilateral side of the lesion did not alter the nociceptive thresholds on the contralateral side [50,51]. MCS could reverse the mechanical hyperalgesia phenomenon in two-thirds of the neuropathic rats, corroborating our group’s previous observations [32,38,52] and those of others [37,53], and partially reversed the allodynia phenomenon. These results emphasize that allodynia is a far more complex disorder, that often requires chronic treatment to provide sustainable antiallodynic effects [36]. Concerning the absence of analgesia, 38% of CCI rats did not respond significantly to MCS, corroborating a previous preclinical study developed by our group [38] and the refractoriness observed in human patients [54,55]. Considering that MCS is often used as a treatment for pharmacological refractoriness to individuals suffering from NP, it is of the utmost importance to better comprehend the differences regarding the responsiveness and refractoriness of MCS. Here, we divided the stimulated animals into two groups according to their responses to MCS (responsive and refractory), in an attempt to highlight the molecular signature of the analgesic response. With that purpose, well established mediators, that are hallmarks of NP, were investigated in the sciatic nerve, DRG, and plasma of the different groups.

NP occurs as a result of peripheral and central sensitization, which is mediated by the activation of glial cells [9,10,56,57], prolonged action of inflammatory cytokines such as TNF-α, IL-1β, IL-6, IL-17, and IFNγ [58,59], and modulation of neurotrophic factors, such as NGF and BDNF, which contribute to nociceptive hypersensitivity and maintenance of NP [56,60,61,62]. Considering that these are well established pain and analgesia mediators, we aimed to understand the peripheral inflammatory profile in MCS-responsive and -refractory animals. We observed that non-treated peripheral neuropathy increased the expression of IL-1β proinflammatory cytokines in the nerve and DRG, as well as decreasing the expression of the anti-inflammatory cytokines IL-4 and IL- 10 in the nerve, corroborating previous studies [39,63,64]. Additionally, CCI decreased IL-17 levels in the nerve, consistent with the findings obtained by Austin et al. [63]. While IL-17 has been reported as a potent proinflammatory cytokine, involved in the development and maintenance of NP [58,65,66,67], it can also be related to the differentiation of macrophages from type M1 (proinflammatory) to type M2 (anti-inflammatory) [68]. Considering the low levels of this cytokine observed in the sciatic nerve of animals with NP, two hypotheses can be suggested: (1) IL-17 contributes to inflammation and pain in the initial phase of nerve damage, and thus it is degraded, or (2) IL-17 acts to inhibit the inflammatory response and consequently the NP. CCI was unable to change the expression of TNF-α, IL-6, IFNγ, CX3CL1, and NGF in the sciatic nerve 14 days after the lesion compared to the control group (FOP), in contrast to findings reported in the literature [40,69,70]. However, the release of NGF, cytokines, and chemokines occurs as an immediate response of the immune system against nerve damage, mediating the recruitment of macrophages in the initial 24 h, which induces the release of algogenic factors that contribute to peripheral sensitization [71,72]. In this sense, it was shown that NGF expression in the sciatic nerve is unchanged 7 days after peripheral nerve injury [73], justifying the absence of its increase 14 days after CCI.

With respect to the sciatic nerve, in a rat model of peripheral neuropathy, MCS was found to induce nerve regeneration and muscle reinnervation, observed by functional and electrophysiological recovery, after sciatic nerve transection followed by microsurgical repair [74]. In this study, we elucidated the physiological effects of MCS on peripheral sensory nerve inflammation induced by peripheral neuropathy. We observed that in the responsive group, IL-10 and IL-17 levels in the sciatic nerve were restored compared to those in the control group. Interestingly, refractory MCS animals did not show the same restoration pattern. Hence, the effectiveness of MCS may be, at least partially, a consequence of an anti-inflammatory action at the injury site, increasing IL-17-induced differentiation of macrophages from M1 to M2, thus contributing to the restoration of IL-10 levels [68]. The anti-inflammatory action of IL-10 occurs due to its selective blocking of the expression of proinflammatory cytokines, chemokines, and cell surface molecules involved in the spread of inflammation [75]. Contributing to the rationale of responsive animals to MCS being related to an increase in the M1 to M2 switch, we observed an enhancement of IL-6 in the sciatic nerve of rats that showed analgesia, which, similar to IL-17, may inhibit proinflammatory cytokine expression [76,77,78,79]. Slight and specific inflammatory inhibition is closely related to the analgesic effect, as the MCS-refractory group showed lower levels of IL-6, IL-10, and IL-17. Additionally, an exaggerated expression of TNF-α was observed in the sciatic nerve of MCS-refractory animals, suggesting pronounced dysregulation of inflammatory factors at the lesion site. The exacerbated increase in proinflammatory factors generates a positive feedback, an uninterrupted cycle of production–stimulation–production, which contributes to the maintenance of high levels of proinflammatory cytokines [79], thus supporting the refractoriness of MCS. Furthermore, in the MCS-responsive group, there was a marked inhibition in NGF expression at the sciatic nerve compared to the control group; this inhibition was not observed in MCS-refractory animals. Corroborating our findings, a decrease in NGF expression at the injury site has been shown in animals with peripheral neuropathy responsive to different therapeutic interventions [40,69,70]. Furthermore, we aimed to investigate if there is a correlation between its expression and the nociceptive threshold, and we observed a high negative correlation between them, showing that MCS-refractory animals have higher expression of this neurotrophin. In this sense, the importance of NGF inhibition in treating persistent pain has been widely discussed in the literature [80,81,82,83]. Considering these findings together, we suggest that, while NP induces a sustained proinflammatory environment in the sciatic nerve, MCS-induced analgesia is accompanied by a slight change in the inflammatory pattern, evidenced by the increase expression of anti-inflammatory cytokines and decreased algogenic factor NGF. This anti-inflammatory role suggests that MCS has a top-down effect that modulates the lesion site, thus decreasing peripheral sensitization and consequently attenuating NP symptoms. Interestingly, MCS-refractory animals failed to induce this switch, suggesting that modulating the inflammatory signature in the lesion site may be pivotal for the analgesic response of MCS.

In regard to DRG, peripheral neuropathy increased the expression of IL-1β in this structure, corroborating data from the literature [84,85]. We did not observe any change in the expression of other cytokines investigated in the DRG after two weeks of lesion induction, which can be explained by the peak release of each cytokine in relation to the time of the analysis [41,86,87,88]. Regardless of its therapeutic effect, MCS reversed the increased levels of IL-1β in the DRG. However, in the refractory group, decreased expression of TNF-α and IL-17 in the ganglia was observed, and this inhibition may be the result of the exacerbated release of TNF-α at the lesion site of animals non-responsive to MCS [89,90]. Despite the absence of any IL-4 cytokine expression change in the DRG, we found a high negative correlation between its expression and the nociceptive threshold in MCS-refractory animals, i.e., refractory animals express higher levels of IL-4 in the DRG. In this sense, anti-inflammatory cytokines, such as IL-4 and IL-10, act to inhibit proinflammatory factors and immune system cells, maintaining the homeostatic balance of the immune response [41,91]. In line with this rationale, and reinforcing our hypothesis, we believe that neuroimmune balance and local control of inflammation are critical for the modulation of NP, and understanding the molecular profile of the MCS-refractory group is necessary for the improvement of this therapy in patients with NP.

In an attempt to find a possible non-invasive MCS-response biomarker in the plasma, we evaluated a neurotransmitter involved with pain (SP) [92,93], and another involved with analgesia (β-endorphin) [42,94]. We observed an increase in SP levels accompanied by a decrease in β-endorphin levels in the plasma of rats with CCI-induced NP. SP is a neuropeptide released by different cell types, promoting a relevant immune response that actively participates in inducing and maintaining the NP [95,96,97,98]; β-endorphin is an endogenous opioid peptide released by local leukocytes at the injury site, and is involved in modulating the transduction and transmission of the nociceptive impulse by inhibiting the release of nociceptive factors, including SP [43,99,100]. Indeed, increased SP expression in the DRG, spinal cord, and plasma levels in NP animals, and its decrease after analgesic treatment, have been demonstrated [44,101,102,103,104]. Here, we observed that MCS, regardless of its nociceptive effect, decreased the plasma levels of SP, suggesting that antinociception or refractoriness is not directly linked to SP levels. In the MCS-responsive group, circulating β-endorphin levels were restored, while only non-responsive MCS animals showed a decrease in plasma β-endorphin levels. It has also been reported that the effectiveness of non-invasive cortical stimulation for pain relief is related to increased endogenous opioid plasma levels in individuals with phantom limb pain [105]. Furthermore, β-endorphins may play a pivotal role in neuroinflammation. In this sense, type M2 macrophages secrete β-endorphin, among other anti-inflammatory agents, which mediates the analgesic response in the CCI model [106]. It has been demonstrated that increased IL-10 in the spinal cord of animals with NP enhances β-endorphin expression in spinal microglia [107]. Considering our findings of local restoration of IL-10 and IL-17 and increased circulating β-endorphin in MCS-responsive animals, it is possible to suggest a relationship between anti-inflammatory compounds and the analgesic-related opioid balance in NP control.

Our data suggest that, considering that NP is a complex, multifactorial disorder of the peripheral and central systems, combining several mechanisms to induce analgesia and modulate several pain mediators is pivotal. Hence, the responsiveness to MCS is a consequence of decreased local proinflammatory cytokines and increased systemic β-endorphin. At the same time, the refractoriness may be related to exacerbated local inflammation, possibly due to a damaged top-down effect of the analgesic system, which cannot increase systemic opioids. In summary, our results demonstrate that NP is accompanied by increased IL-1β in the sciatic nerve and DRG, inhibition of IL-4, IL-10, and IL-17 in the nerve, enhanced SP, and reduced β-endorphin in the plasma, compared to the control group. While MCS reduces pain behavior, a percentage of animals do not respond to this technique, and regardless of its therapeutic effect, the stimulation inhibits IL-1β in the DRG and SP in the plasma. In MCS-responsive animals, there was an increase in IL-6, restoration of IL-10 and IL-17 levels, a decrease in NGF in the nerve, and an increase in plasma β-endorphin, attenuating CCI-induced NP. In refractory animals, there was an increase in TNF-α and a decrease in IFNγ in the nerve, and a decrease in TNF-α and IL-17 in the DRG (Figure 6).

As a final remark, it is difficult to propose a direct translational comparison, as seen in every preclinical study, however, in line with the recent literature [50], it is important and possible to trace some response predictors in humans, peripheral and central biomarkers, that may help to improve the MCS responsiveness. So, a better understanding of the mechanisms involved in the analgesic effect of MCS in animals with persistent NP may contribute to the general understanding of the technique, thus guiding a more significant benefit of this therapeutic intervention for patients with NP refractory to other conventional interventions.

## 4. Materials and Methods

### 4.1. Animals

Forty-six male Wistar rats, weighing 180–220 g, obtained from the University of São Paulo, were used as experimental subjects. Animals were housed in polypropylene cages (40 cm × 34 cm × 17 cm) with wood shavings (three rats/cage) for at least 1 week before the beginning of the experiments. In an appropriate room, with controlled ambient temperature (22 ± 2 °C) and a 12/12 h light/dark cycle, the animals received water and rat chow pellets ad libitum. All experimental procedures were conducted in strict adherence to the animal research reporting of in vivo experiments (ARRIVE) guidelines [108] and the guidelines for the ethical use of animals in research involving pain and nociception [109]. This study was approved by the Ethics Committee on the Use of Animals at Hospital Sírio-Libanês (Protocol No. CEUA 2014-04).

### 4.2. Induction of Neuropathic Pain

Animals underwent CCI of the sciatic nerve in the right paw, according to Bennett and Xie [110] (Figure 1), under general inhalation anesthesia with isoflurane (4% isoflurane in 100% oxygen to induce anesthesia, and 2.5% and 100% oxygen to maintain it). In this procedure, the sciatic nerve was exposed in the median region of the thigh, away from the biceps femoris muscle, and four loose ligatures (chrome-plated catgut 4-0) were made around it, approximately 1 mm apart. The incision was sutured using a silk suture (4-0). The sciatic nerve of FOP rats was surgically exposed without compression. The animals were monitored during the postoperative period until complete recovery from anesthesia and for up to 48 h following surgery.

### 4.3. Electrode Implantation and MCS

After one week of CCI or FOP, the animals again received general anesthesia with isoflurane, associated with local anesthesia (2% lidocaine, 100 μL/animal on the scalp). Under stereotaxic conditions, a pair of transdural electrodes (cylindrical stainless steel with 0.8 mm diameter) was inserted over the primary motor cortex area corresponding to the right posterior limb (1.5 mm posterior to the bregma and 1 mm anterior to bregma, both inserted 1 mm from the median line) (Figure 1), according to a functional map of the rat motor cortex previously constructed by our group [47]. Two fixation screws were implanted on the left side. The electrodes were attached to the skull using acrylic polymerizing resin, and the electrode extensions were inserted into a socket for further connection to the stimulator cables, fixed with acrylic polymer, which also works as an electrical insulator for the socket contacts. One week after implantation of the electrodes, the animals underwent a single session of MCS, performed for 15 min (1 V, current between 0.9 and 1 mA; 60 Hz; 210 μs of pulse duration; Medtronic electrical stimulator, model 3625, Minneapolis, MN, USA), according to the standardization performed previously [111]. At the end of the experiments, all electrodes were tested to ensure an electric function using a digital multimeter apparatus.

### 4.4. Nociceptive Sensitivity Evaluation

Nociceptive tests were performed before any surgical intervention (initial measurement, IM) and on the 14th day after CCI or FOP, before (intermediate measurement, IntM) and after 15 min, with or without cortical stimulation (final measurement, FM). The results were analyzed by comparing the initial and final measurements between all experimental groups.

#### 4.4.1. Mechanical Hyperalgesia Assessment

Mechanical hyperalgesia was assessed using the paw pressure test (EEF 440, Analgesimeter, Insight^®^, Ribeirão Preto, SP, Brazil) [112]. A force, in grams, of increasing intensity was continuously applied to the back of the right hind paw, and its nociceptive response was determined based on the reaction of withdrawing, thus indicating the nociceptive threshold of each animal.

#### 4.4.2. Mechanical Allodynia Assessment

Allodynia was evaluated using von Frey filaments (Stoelting^®^, Wood Dale, IL, USA) in response to a tactile stimulus applied for 8 s to the plantar area of the right posterior paw [113]. Animals were habituated for 30 min to the test during the two days preceding the experiments and on the experimental day for 15 min. A logarithmic series of seven filaments was used, which started with 4.56 (3.63 g), and according to the animal’s response (whether or not it withdrew the paw), a thinner or thicker filament was applied, following the sequence: 3.61 (0.41 g), 3.84 (0.70 g), 4.17 (1.50 g), 4.56 (3.63 g), 4.93 (8.5 g), 5.18 (15.10 g), and 5.46 (28.9 g). To analyze the animal’s nociceptive threshold, six responses were collected, to predict the final behavior in grams.

### 4.5. Tissue and Plasma Sample Collection

On day 14° after the nociceptive tests, the animals were immediately euthanized with a guillotine, and the right sciatic nerve and right DRG (L4, L5, and L6 portion) were collected, frozen in liquid nitrogen, and stored at −80 °C. Blood was collected through cardiac puncture. It was collected in an EDTA tube, preserved on ice for a maximum of 2 h, and centrifuged at 800× *g* for 10 min at 4 °C. Subsequently, the plasma was separated and centrifuged again at 1600× *g* for 10 min at 4 °C, aliquoted, and frozen at −80 °C.

### 4.6. Protein Extraction

Protein extraction was performed on samples of the sciatic nerve and DRG (L4, L5, and L6 pool) using lysis buffer (137 mM NaCl, 20 mM Tris-HCl (pH 8.0), 1% NP40, 10% glycerol, 1 mM PMSF, 10 μg/mL aprotinin, 1 μg/mL leupeptin, 0.5 mM sodium vanadate). Each analyzed tissue sample was standardized using an ideal volume of extraction buffer. The total protein content was measured using the Pierce^®^ (Thermo Fisher Scientific, Waltham, MA, USA) method, with an Infinite^®^ M200 PRO reader (Tecan, Mannedorf, ZRH, CH). The absorbance was read at 660 nm.

### 4.7. Enzyme-Linked Immunosorbent Assay (ELISA)

The neurotrophins NGF (DY556, R&D, Minneapolis, MN, USA) in the sciatic nerve and plasma, and BDNF (G7610, Promega, Madison, WI, USA) in the plasma, were quantified by an immunoenzymatic assay (ELISA) following the manufacturer’s recommendations.

### 4.8. Multiplex Assay

Animal samples were evaluated using a rat cytokine/chemokine panel (RECYTMAG-65K, Merck Millipore, Burlington, MA, USA) for TNF-α, IL-1β, IL-4, IL-6, IL-10, IL-17A, IFNγ, and CX3CL1, in the sciatic nerve and DRG, and by the rat neuropeptide panel (RECYTMAG-83K, Merck Millipore) for SP and β-endorphin in the plasma, using a commercially available multiplex magnetic bead-based kit (Magpix^®^, Luminex Corp., Austin, TX, USA). Tests were performed according to the manufacturer’s recommendations.

### 4.9. Statistical Analysis

The sample size of the animals was established by considering the paw pressure test as the primary outcome [114], where the power (β) was 80%. Results are expressed as the mean ± SEM. Data were analyzed with GraphPad Prism 5 (GraphPad Software Inc., Boston, MA, USA), and statistical significance was assessed using two-way analysis of variance (ANOVA) (2-w-ANOVA), followed by Bonferroni’s multiple comparison post hoc tests for behavioral measurements. For the analysis of the ELISA and multiplex assays, one-way ANOVA was used (compared groups: FOP × CCI × MCS), followed by Tukey’s multiple comparison post hoc tests. Pearson’s test was applied for correlation analysis, considering mean correlation between r^2^ = 0.5 to 0.75 and high correlation when r^2^ > 0.75. Low correlations (r^2^ < 0.5) were not considered.

## 5. Conclusions

MCS reverses mechanical hypernociception, at least in part, owing to the modulation of cytokines and NGF at the injury site and an increase in circulating opioids. Refractoriness to MCS involves exacerbated inflammation, seen by an increased level of TNF-α and low levels of IL-10, IL-17, and IFNγ at the lesion site, in addition to a decrease in IL-17 in the DRG and a deficit in the opioid system, as seen by plasma β-endorphin.

## Figures and Tables

**Figure 1 ijms-24-07796-f001:**
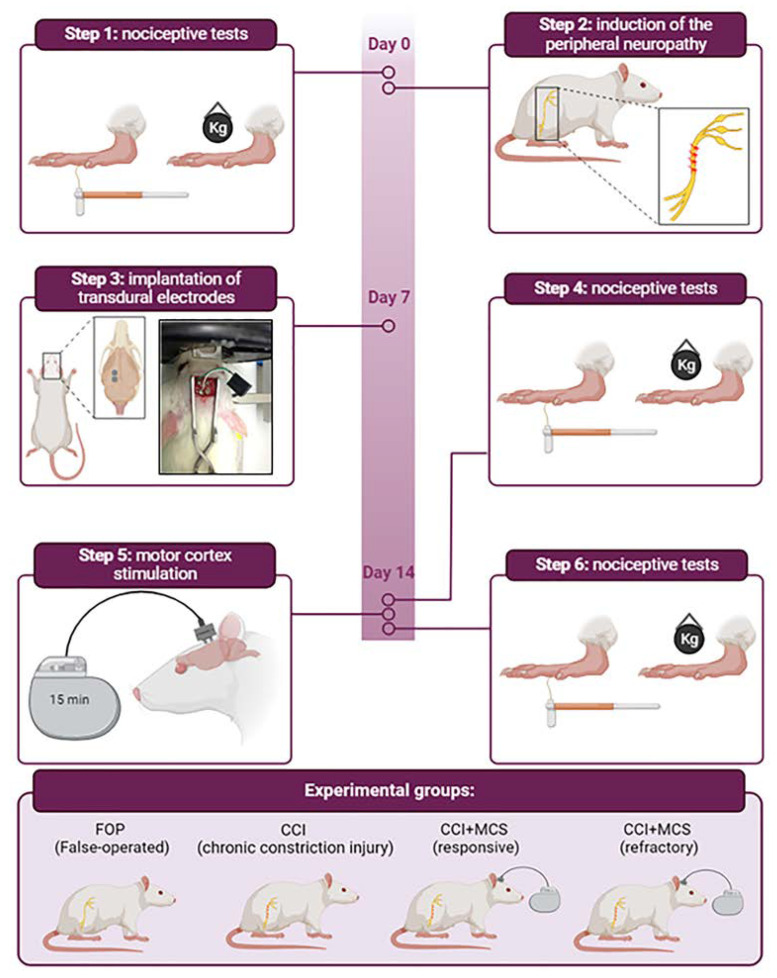
Experimental design, CCI induction, and MCS procedure. Temporal scheme of the procedures performed with the animals during the 14 days of experimentation. The timeline description: Day 0 represents the basal measurement of the nociceptive tests. Animals were submitted to the chronic constriction injury (CCI) of the right sciatic nerve, and false-operated animals (FOP) were used as control. Day 7, two transdural electrodes were placed on the primary motor cortex over the right hind limb area. Day 14, animals were initially divided into three experimental groups: FOP, CCI, and CCI + MCS. They were evaluated using nociceptive tests. CCI + MCS animals were submitted to 15 min of MCS and, still under stimulation, were reevaluated in the tests. After this 15 min of MCS, the third group was subdivided into CCI + MCS responsive or CCI + MCS refractory, totaling four experimental groups: FOP, CCI, CCI + MCS responsive and CCI + MCS refractory. FOP and unstimulated CCI animals were also reevaluated in the tests. After the last test, animals were euthanized, and the tissues were subjected to different assays. IM, initial measurement; intM, intermediate measurement; FM, final measurement. Adapted from “multi-panel vertical timeline” by BioRender.com, accessed on 1 September 2022.

**Figure 2 ijms-24-07796-f002:**
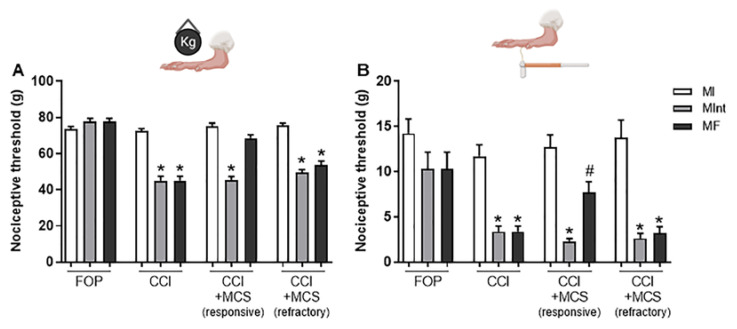
Effect of MCS treatment on the hypernociception of animals with peripheral neuropathy. Animals were evaluated in the paw pressure test (**A**) and with von Frey filaments (**B**), to detect the mechanical hyperalgesia and allodynia, respectively, in the right paw, before any surgical intervention (initial measurement, IM), after 14 days of FOP or CCI (intermediate measurement, IntM) and 15 min after MCS, still under stimulation (final measurement, FM). FOP and unstimulated CCI animals were also evaluated (*n* = 10 animals/group). * *p* < 0.001 in relation to MI; # *p* < 0.05 in relation to IM and IntM. FOP, false-operated; CCI, chronic sciatic nerve constriction; CCI + MCS, stimulation of the motor cortex in animals with CCI.

**Figure 3 ijms-24-07796-f003:**
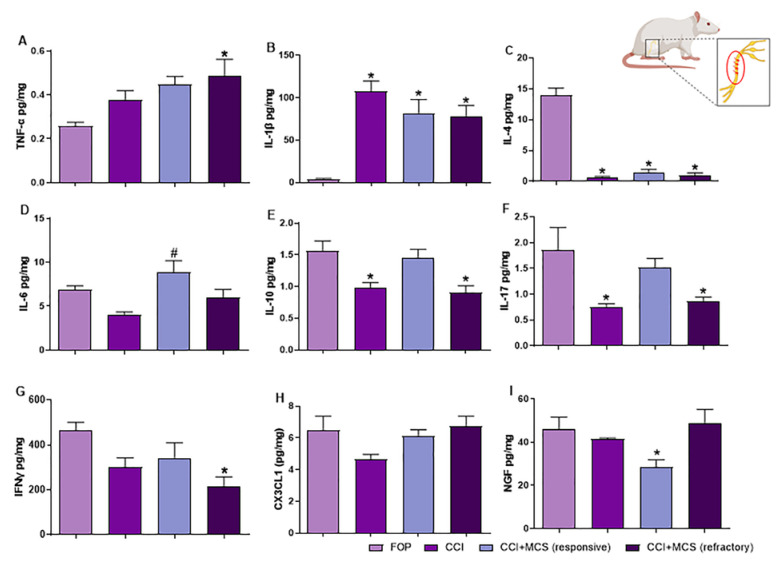
Effect of MCS treatment on inflammatory and neurotrophic profile in the sciatic nerve of rats with peripheral neuropathy. Quantitative analysis of multiplex assay for TNF-α (**A**), IL-1β (**B**), IL-4 (**C**), IL-6 (**D**), IL-10 (**E**), IL-17 (**F**), IFNγ (**G**), CX3CL1 (**H**), and NGF (**I**) on the sciatic nerve of false-operated (FOP) rats, with peripheral neuropathy (CCI), with CCI responsive to MCS (CCI + MCS responsive), and with CCI refractory to MCS (CCI + MCS refractory) (*n* = 5 animals/group). * *p* < 0.05 compared to the FOP group. # *p* < 0.05 compared to the CCI group.

**Figure 4 ijms-24-07796-f004:**
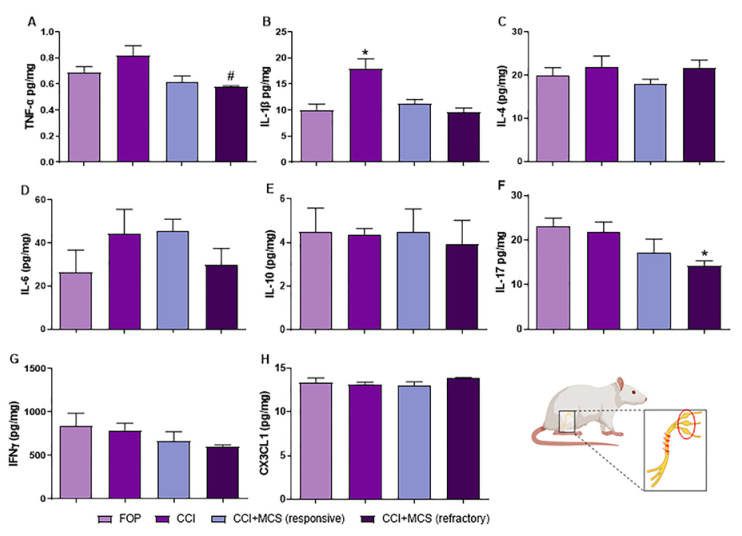
Effect of MCS treatment on the inflammatory and neurotrophic profile in the DRG of rats with peripheral neuropathy. Quantitative analysis of multiplex assay for TNF-α (**A**), IL-1β (**B**), IL-4 (**C**), IL-6 (**D**), IL-10 (**E**), IL-17 (**F**), IFNγ (**G**), and CX3CL1 (**H**) on the sciatic nerve of false-operated (FOP) rats, with peripheral neuropathy (CCI), with CCI responsive to MCS (CCI + MCS responsive), and with CCI refractory to MCS (CCI + MCS refractory) (*n* = 5 animals/group). * *p* < 0.05 compared to the FOP group. # *p* < 0.05 compared to the CCI group.

**Figure 5 ijms-24-07796-f005:**
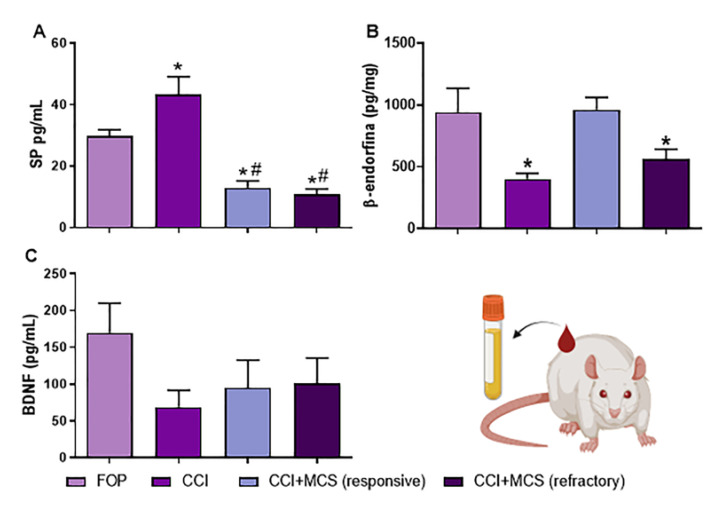
Pattern of circulating plasma biomarkers after MCS treatment. Expression analysis of SP (**A**), β-endorphin (**B**), and BDNF (**C**) on the plasma of false-operated rats (FOP), with peripheral neuropathy (CCI), with CCI responsive to MCS (CCI + MCS responsive), and with CCI refractory to MCS (CCI + MCS refractory) (*n* = 5 animals/group). * *p* < 0.05 compared to the FOP group. # *p* < 0.05 compared to the FOP and CCI group.

**Figure 6 ijms-24-07796-f006:**
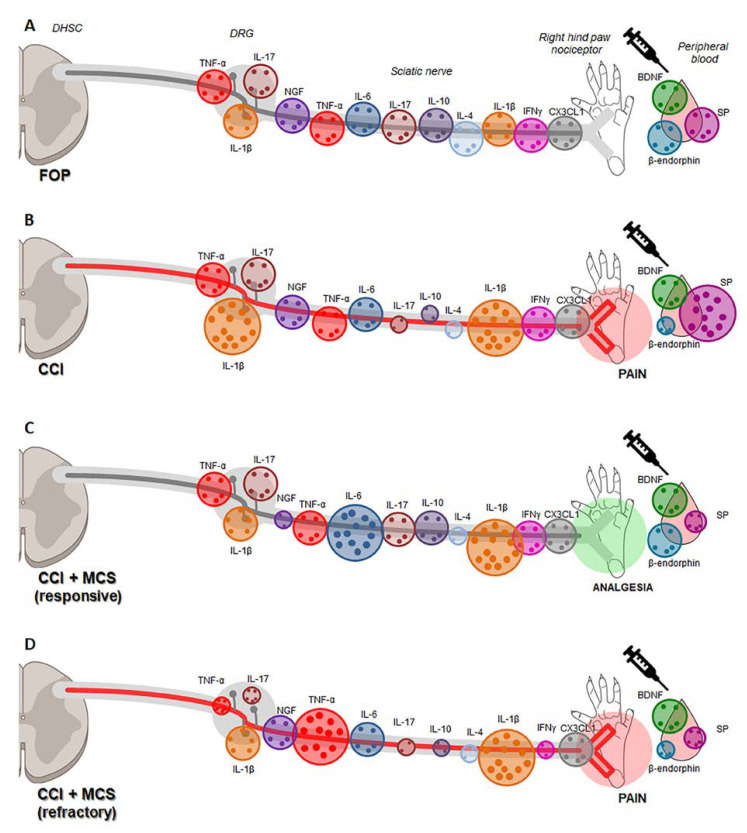
Representative illustration for the differential expression of factors involved in NP control against MCS responsiveness and refractoriness in rats. Didactic illustration of cytokines TNF-α, IL-1β, IL-4, IL-6, IL-10, IL-17A, IFNγ, and neurotrophin NGF expression in the sciatic nerve and DRG of false-operated (FOP, **A**) rats, with peripheral neuropathy (CCI, **B**), with CCI responsive to MCS (CCI + MCS responsive, **C**), and with CCI refractory to MCS (CCI + MCS refractory, **D**), as well as SP, β-endorphin, and BDNF expression in plasma of the same groups. DHSC: dorsal horn of the spinal cord.

**Table 1 ijms-24-07796-t001:** Correlation between nociceptive threshold and cytokine, neurotrophin, β-endorphin, and SP expression in different animal structures, comparing based on the MCS responsiveness.

**Sciatic Nerve**		
**Analytes**	**Correlation Between Groups**	**Results**
TNF-α × nociceptive threshold	Responsive × Refractory	r^2^ = 0.22 *p* = 0.12
IL-1β × nociceptive threshold	Responsive × Refractory	r^2^ = 0.05 *p* = 0.47
IL-4 × nociceptive threshold	Responsive × Refractory	r^2^ = 0.01 *p* = 0.71
IL-6 × nociceptive threshold	Responsive × Refractory	r^2^ = 0.15 *p* = 0.16
IL-10 × nociceptive threshold	Responsive × Refractory	r^2^ = 0.46 *p* = 0.2
IL-17 × nociceptive threshold	Responsive × Refractory	r^2^ = 0.40 *p* = 0.1
IFNγ × nociceptive threshold	Responsive × Refractory	r^2^ = 0.10 *p* = 0.30
CX3CL1 × nociceptive threshold	Responsive × Refractory	r^2^ = 0.14 *p* = 0.23
NGF × nociceptive threshold	Responsive × Refractory	r^2^ = 0.89 *p* = 0.004
**DRG**		
**Analytes**	**Correlation Between Groups**	**Results**
TNF-α × nociceptive threshold	Responsive × Refractory	r^2^ = 0.03 *p* = 0.71
IL1-β × nociceptive threshold	Responsive × Refractory	r^2^ = 0.08 *p* = 0.58
IL-4 × nociceptive threshold	Responsive × Refractory	r^2^ = 0.77 *p* = 0.02
IL-6 × nociceptive threshold	Responsive × Refractory	r^2^ = 0.22 *p* = 0.52
IL-10 × nociceptive threshold	Responsive × Refractory	r^2^ = 0.02 *p* = 0.76
IL-17 × nociceptive threshold	Responsive × Refractory	r^2^ = 0.05 *p* = 0.64
IFNγ × nociceptive threshold	Responsive × Refractory	r^2^ = 0.03 *p* = 0.71
CX3CL1 × nociceptive threshold	Responsive × Refractory	r^2^ = 0.41 *p* = 0.16
**Plasma**		
**Analytes**	**Correlation Between Groups**	**Results**
β-endorphin × nociceptive threshold	Responsive × Refractory	r^2^ = 0.37 *p* = 0.05
SP × nociceptive threshold	Responsive × Refractory	r^2^ = 0.09 *p* = 0.32
BDNF × nociceptive threshold	Responsive × Refractory	r^2^ = 0.03 *p* = 0.64

## Data Availability

All other data generated and analyzed are either included in this published article or available from the corresponding author upon reasonable request.

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
