# Peer review of "Systemic and Peripheral Mechanisms of Cortical Stimulation-Induced Analgesia and Refractoriness in a Rat Model of Neuropathic Pain"

_ijms, 2023, doi:10.3390/ijms24097796_

Round 1

Author Response

The manuscript was reviewed according to the comments and the changes/answers are listed below. All changes are highlighted in yellow in the reviewed manuscript.

REVIEWER 1

Comments:

The manuscript (ijms-2242425) entitled “Systemic and peripheral mechanisms of cortical stimulation-induced analgesia and refractoriness in a rat model of neuropathic pain” seems to be interesting initially, however, the authors tried to study the neuropathic pain CCI and motor cortex stimulation’s responsiveness-refractoriness. The hypothesis proposed in this manuscript is vague, fuzzy and lack clear scientific conclusion. The experimental designs lack novelty, and the writing must be improved at higher degree at scientific level. The result concluded with deficit in scientific justification.

Major Issues:

Reviewer: The neuropathic pain rat model in the field of pain management is well established. Author evaluates the inflammatory cytokines, neurotrophins, Subastance P, β-endorphin in different groups FOP, CCI, CCI+MCS (responsive & refractory) and nociceptive test, which is routine, no novelty.

Response: We totally agree that the neuropathic pain rat model used in this project and in the field of pain management is well established. Because the aim of this project is to elucidate the main differences between refractoriness and responsiveness of motor cortex stimulation, we decided to use a reliable and well stablished model that would minimize the possible confounders of a less explored model. Therefore, the novelty of our experimental design with the proposed groups is to study and try to better understand the refractoriness after cortical stimulation. The FOP and CCI animals’ group were used as an experimental control of the absence or presence of pain, respectively. These data we consider  of  most importance  as  a  negative/positive control of the methodology, not only of pain evaluation, but also for the classic mediators chosen. In this sense, the results obtain from the stimulated groups (that induced or not analgesia) can be related to the proposed treatment.

In a similar manner, we aimed to investigate well stablished pain mediators that have been well described in the literature in regards of its role in neuropathic pain in the attempt to unravel the ones that may be related to the refractoriness or responsiveness of MCS. We are very sorry for not making this information clearer. We added this rationale in the reviewed manuscript (lines 74-78).

Reviewer: Line 105; As per the diagrammatic representation four groups given, confirm 3 or 4 group justify.

Response: We are sorry for the misleading information. In a better explanation, initially the animals were divided into 3 experimental groups (FOP, CCI and CCI+MCS) and at the end of the 15 minutes of stimulation, we obtained another division, the responsive and refractory stimulated group, thus totaling 4 experimental groups: FOP, CCI, CCI+MCS responsive and CCI+MCS refractory). This information can be found in line 107 in the reviewed manuscript.

Reviewer: Result section for expression pattern not explained properly for figure 3-5, requires more scientific interpretation.

Response: We appreciate the input of the reviewer. As such, we add a reminder of the importance of these mediators in pain response for each section, with the paragraph “Considering that NP is closely related to peripheral neuroinflammation, …”, in the beginning of the 2.3 topic (line 139). Also, we added the paragraph “To continue the investigation of peripheral neuroinflammation, we evaluated cytokines, classically involved in persistent pain [references], in the neuronal bodies cluster found in the DRG” in 2.4 topic (line 167) and the paragraph “For a deeper investigation, we analyzed the circulant levels of factors involved in the NP control [references] and interestingly…”, in 2.5 topic (line 190). Additionally, the reviewer can find further scientific interpretation of the results obtained in the Discussion section in the reviewed manuscript.   

Reviewer: The results section needs better explanation with inter-correlation predictions within the group and with the Sciatic and DRG. Required more specific and effective elaboration of result. The NGF was not quantitated in DRG but claimed neurotrophic factor assessed in abstract. Appreciates the author's diagrammatic representation of result for better reach of outcome.

Response: We appreciate all the comments and alterations proposed, and we tried to contemplate all of them. In relation to the to the first point raised by the reviewer, some changes were done and highlighted, trying to better explain our results during the text. Furthermore, a linear correlation between nociceptive threshold and the mediators studied in the sciatic and DRG was added in a new topic “2.6. Correlations between nociceptive threshold, expression of cytokines, neurotrophins, β-endorphin or SP with MCS” (lines 207-217). Also, a table with these results of correlation was added in the Table 1. The findings of correlation were also discussed in the Discussion section (lines 323-325 and 344-349) in the reviewed manuscript.

Lastly, with regard to NGF not quantitated in DRG but claimed neurotrophic factor assessed in abstract, we think that it is a misunderstood, because the sentence in the abstract concerning to this analysis is “The expression of cytokines in the dorsal root ganglion (DRG) and sciatic nerve, NGF in the nerve, and NGF, BDNF, SP, and β-endorphin in the plasma was analyzed”. In another words we have analyzed in the DRG: cytokines; in the sciatic nerve: cytokines and NGF; and in the plasma: NGF, BDNF, β-endorphin and SP. We deeply apologize for the mistake in the Results section, topic title 2.4. “Expression pattern of cytokines and neurotrophins in the DRG”. We changed this title to "Expression pattern of cytokines in the DRG” (line 166) in the reviewed manuscript.

Reviewer: The discussion section is very fuzzy, no proper correlative discussion makes more confusable. Required to rewrite the discussion in clear with effective flow of result discussion with previous published data to hold attention of reader.

Response: We appreciate your notes and have made some changes to the writing in order to clarify and maintain a clear effective thinking flow.

Minor issues:

Reviewer: Line 365; 2.5% isoflurane infused with oxygen during whole surgery procedure? and is that any use of thermostatic heat pad used for maintenance of rat body temperature??

Response: Thanks for your observation. Yes, animals were at first submitted to 4% isoflurane in 100% oxygen to induce anesthesia. Next, 2.5% isoflurane was infused with 100% oxygen during the whole surgery procedure. This information was added in the Material and Methods section (line 425). Regarding the second point raised by the reviewer, we followed the standard anesthesia protocol approved by the animal ethics committee of our institution to insure body temperature and analgesia.

Reviewer: Line 386; mention MCS instrument make, and model used.

Response: We apologize for the absence of this information in the text. We added the make and model in the Material and Methods section (line 447) in the reviewed manuscript.

Reviewer: Line 418; route of blood collection needs to mention.

Response: We absolutely agree with the reviewer and are very sorry for not disclosing this information before. We added this information in the Material and Methods section (line 480) in the reviewed manuscript.

Reviewer: Line 429; Mention kit catalogue number and absorbance read in the instrument.

Response: Thank you for your observation. We have included the request information in the Material and Methods section (lines 490-492) in the reviewed manuscript.

Reviewer: Line 433; Result of Plasma NGF needs to mention and rewrite the result section for NGF.

Response: The results concerning NGF in the plasma were given in Results section, topic 2.5; however, we apologize for the absence of the word NGF in the title of this section. To clarify this information, we have changed the title of topic to “Expression pattern of SP, β-endorphin, BDNF and NGF in the plasma”, in the Results section (line 189), in the reviewed manuscript.

Reviewer: Line 458; Section 6 needs to add the content such as patent number with brief description.

Response: This project did not generate any patent, so we removed this topic in the reviewed manuscript.

Reviewer: Overall, the manuscript lacks a clear scientific hypothesis, novel experimental designs and concluded in improperly. The entire study seems to be descriptive and must improve far better before considering for publication.

Response: We appreciate the reviewer’s concern. The rationale of this work is that the failure of MCS, which causes deterioration of quality of life of individuals with neuropathic pain, may be attenuated once we unravel the main differences between the effectiveness and refractoriness of this neuromodulation technique. As such, the novelty of our experimental design does not rely in the study of pain per se. The use of well stablished and worldwide preclinical model, as well as the classic mediators involved in neuropathic pain were carefully chosen with the purpose of better comprehending the aim of this work, which is the treatment response of MCS. We are very sorry if this was unclear in our manuscript. Therefore, we added our aim and hypothesis in the end of Introduction section (lines 74-78). To the best of our knowledge this is the first work in the literature that extendedly investigates the main differences between MCS-induced analgesia and failure. Considering the complexity of persistent pain, rather than investigating the effect of MCS-induced analgesia and blocking or attenuating the response of a single mediator, we aimed to fully comprehend the many responses that MCS fails to induce resulting in the refractoriness of this neuromodulation technique.

Reviewer 2 Report

This is the first really deeply scientifically provided evidence on the effectiveness of epidural motor cortex stimulation (MCS) for neuropathic pain relief in experimental conditions. It’s difficult to ask rats about pain relief, but the authors provide convincing evidence on the refractoriness of pain with the expression of cytokines, neurotrophins and nociceptive mediators in rats with peripheral neuropathy. They concluded that the refractory animals showed increased TNF-α and decreased IFNγ in the nerve, along with decreased TNF-α and IL-17A in the DRG, maintaining the CCI-induced low levels of systemic β-endorphin. Their findings convincingly suggest that MCS effectiveness depends on local control of inflammatory and neurotrophic changes, accompanied by recovery of the opioidergic system deficit observed in neuropathic conditions. They answered the main questions included in the aim, not to the end, but interestingly.

The article scientifically sounds like a real piece of hard work.  I have found their intentions very positive.

Point by point please:

Some major revisions/supplementations should be made which do not exclude this interesting paper for further qualification to be printed in the IJMS:

I know that promises from the basic studies for purposes of clinical applications sometimes sound false, but please promise something at the end of the Abstract for the future, as well as something broader in the Discussion section.

Work harder for the editorial part of the article (lines 42,43). Brackets round or square, please decide (line 44).

Lines 76-79, consider the content again; poorly understood, it is the Aim. Rewrite the aim in lines 72-79 as the one paragraph again to be clearly understood. Only the expert understands its content.

“Adapted from “multi-panel vertical  timeline” by BioRender.com (2022).” Line 109 sounds mysterious; what does it mean for readers?

Moreover, however:

Introduction section – perfect

Material and methods section description – No, no, no, where is it? Maybe it is a misunderstanding. Section 2 - Materials and methods should be instead of Results?

Either lack of M&M section, or you miss it at all. Please clarify the rules of your investigation in the Materials and Methods section and separate them from the Results section.

And after the Discussion, which is very interesting and informative, the 4. Material and Methods section appears!

I have read the very interesting and informative Discussion, and I congratulate the Authors on this very interesting paper. I hope they will correct the sequence of the parts of this article again in the correct order.

Rewrite the Conclusions. This is not the Abstract. They in the present form, do not reflect the content of the results. Provide the Key points of your study.

Refs. - Make years bold and names of the journals italics; what would make your edition closer to the MDPI requirements 

I’ve got the feeling that the authors made a really great work, but at the stage of submission, they lost the nerves, which should not be made their work poor, but it really is not.

Author Response

REVIEWER 2

Coments:

This is the first really deeply scientifically provided evidence on the effectiveness of epidural motor cortex stimulation (MCS) for neuropathic pain relief in experimental conditions. It’s difficult to ask rats about pain relief, but the authors provide convincing evidence on the refractoriness of pain with the expression of cytokines, neurotrophins and nociceptive mediators in rats with peripheral neuropathy. They concluded that the refractory animals showed increased TNF-α and decreased IFNγ in the nerve, along with decreased TNF-α and IL-17A in the DRG, maintaining the CCI-induced low levels of systemic β-endorphin. Their findings convincingly suggest that MCS effectiveness depends on local control of inflammatory and neurotrophic changes, accompanied by recovery of the opioidergic system deficit observed in neuropathic conditions. They answered the main questions included in the aim, not to the end, but interestingly.

The article scientifically sounds like a real piece of hard work.  I have found their intentions very positive.

Point by point please:

Reviewer: Some major revisions/supplementations should be made which do not exclude this interesting paper for further qualification to be printed in the IJMS:

I know that promises from the basic studies for purposes of clinical applications sometimes sound false, but please promise something at the end of the Abstract for the future, as well as something broader in the Discussion section.

Response: We appreciate the concern raised by the reviewer, and we changed some point in the middle of the abstract and included a paragraph in the end of it, to give light to the future as mentioned. And to finalize, we have made a close comparison and once again brought to light an improvement of this technique that is so useful and promising for NP treatment by adding this paragraph at line 393: “As a final remark, it is difficult to propose a direct translational comparison, as seen in every preclinical study, however in line with the recent literature [reference], is important and possible to trace some response predictors in humans, peripheral and central biomarkers, that may help to improve the MCS-responsiveness”.

Reviewer: Work harder for the editorial part of the article (lines 42,43). Brackets round or square, please decide (line 44).

Response: Thank you for your observation. The brackets are all square now as the editorial rule.

Reviewer: Lines 76-79, consider the content again; poorly understood, it is the Aim. Rewrite the aim in lines 72-79 as the one paragraph again to be clearly understood. Only the expert understands its content.

Response: We appreciate your point of view. In the initial lines we just justify why is so important a deeper understand of MCS-induced analgesia related to peripheral participation, and also the refractoriness evaluating peripheral mediators. We rewrote the aim in the revised manuscript in an attempt to clarify our proposed study (lines 74-82).

Reviewer: “Adapted from “multi-panel vertical timeline” by BioRender.com (2022).” Line 109 sounds mysterious; what does it mean for readers?

Response: This “Adapted from “multi-panel vertical timeline” by BioRender.com (2022)" is the way we need to indicate where and how the figure was made, so in another word, our timeline was made using a vertical multi-panel by BioRender.

Reviewer: Moreover, however:

Introduction section – perfect

Response: We are grateful for your comment. Thank you so much!

Reviewer: Material and methods section description – No, no, no, where is it? Maybe it is a misunderstanding. Section 2 - Materials and methods should be instead of Results?

Either lack of M&M section, or you miss it at all. Please clarify the rules of your investigation in the Materials and Methods section and separate them from the Results section.

And after the Discussion, which is very interesting and informative, the 4. Material and Methods section appears!

I have read the very interesting and informative Discussion, and I congratulate the Authors on this very interesting paper. I hope they will correct the sequence of the parts of this article again in the correct order.

Response: We greatly appreciate the concern of the reviewer, and would like to better explain. The preparation of the article sections is in accordance with the general considerations of manuscript preparation, Research manuscripts should comprise in the following order: Introduction, Results, Discussion, Materials and Methods, Conclusions (which is optional).

Reviewer: Rewrite the Conclusions. This is not the Abstract. They in the present form, do not reflect the content of the results. Provide the Key points of your study.

Response: We totally agree with the reviewer and rewrote the Conclusion section (lines 520-524) as the following paragraph:MCS reverses mechanical hypernociception, at least in part, owing to the modulation of cytokines and NGF at the injury site and an increase in circulating opioids. Refractoriness to MCS involves exacerbated inflammation seem by an increased level of TNF-α and low level of IL-10, IL-17 and IFNγ at the lesion site, added to a decrease in IL-17 in the DRG and a deficit in the opioid system, as seen by plasma β-endorphin.”

Reviewer: Refs. - Make years bold and names of the journals italics; what would make your edition closer to the MDPI requirements 

Response: We value the comment and have changed at the manuscript references, the years in bold and names of the journals in italic.

Reviewer:I’ve got the feeling that the authors made a really great work, but at the stage of submission, they lost the nerves, which should not be made their work poor, but it really is not.

Response: We are very sorry for this impression. We have carefully reviewed the manuscript and the journal`s guidelines and hope that this will compensate for the initially wrong impression of the reviewer.

Reviewer 3 Report

ijms-2242425: “Systemic and peripheral mechanisms of cortical stimulation-induced analgesia and refractoriness in a rat model of neuropathic pain”.

In this comprehensive experimental study of central control of refractory neuropathic pain, the authors demonstrate that the effectiveness of epidural motor cortex stimulation depends on local control of inflammatory and neurotrophic changes, in parallel with recovery of the opioidergic system functioning. The material is described successively, cogently and intelligibly. This manuscript could be recommended for publication after minor technical corrections:

a)    line 458, “6. Patent” should be separated from previous paragraph;

b)    line 454 and below, duplicated numerations should be omitted.

Author Response

REVIEWER 3

Coments:

In this comprehensive experimental study of central control of refractory neuropathic pain, the authors demonstrate that the effectiveness of epidural motor cortex stimulation depends on local control of inflammatory and neurotrophic changes, in parallel with recovery of the opioidergic system functioning. The material is described successively, cogently and intelligibly. This manuscript could be recommended for publication after minor technical corrections:

 a) line 458, “ 6. Patent” should be separated from previous paragraph;

Response: We are very sorry for this. This project did not generate any patent, so we removed this topic in the reviewed manuscript.

b) line 454 and below, duplicated numerations should be omitted.

Response: Thank you for the point, we absolutely agree with comment and have corrected all the duplicated numbers at the References section.

Round 2

Reviewer 1 Report

No comments.

Author Response

We would like to thank you for the final analysis of the manuscript and the acceptance. We appreciate all the comments and suggestions that have considerably improved our paper to be published in the IJMS. 

Reviewer 2 Report

Considering the content of the IJMS template which presents the rules of the manuscript presentation by MDPI and the authors' explanations and responses to my remarks and questions,  I suggest accepting the manuscript in its current form.

Author Response

(The authors gave the same response as above.)
